# A comparative genomics study of 23 *Aspergillus* species from section *Flavi*

Inge Kjærbølling[1], Tammi Vesth[1], Jens C. Frisvad [1], Jane L. Nybo[1], Sebastian Theobald[1], Sara Kildgaard[1], Thomas Isbrandt Petersen[1], Alan Kuo[2], Atsushi Sato[3], Ellen K. Lyhne[1], Martin E. Kogle[1], Ad Wiebenga[4], Roland S. Kun[4], Ronnie J.M. Lubbers[4], Miia R. Mäkelä [5], Kerrie Barry[2], Mansi Chovatia[2], Alicia Clum[2], Chris Daum[2], Sajeet Haridas [2], Guifen He[2], Kurt LaButti [2], Anna Lipzen[2], Stephen Mondo[2], Jasmyn Pangilinan[2], Robert Riley[2], Asaf Salamov[2], Blake A. Simmons [6], Jon K. Magnuson [6], Bernard Henrissat[7], Uffe H. Mortensen [1], Thomas O. Larsen [1], Ronald P. de Vries [4], Igor V. Grigoriev [2,8], Masayuki Machida[9], Scott E. Baker [6,10] & Mikael R. Andersen [1]*

Section *Flavi* encompasses both harmful and beneficial *Aspergillus* species, such as *Aspergillus oryzae*, used in food fermentation and enzyme production, and *Aspergillus flavus*, food spoiler and mycotoxin producer. Here, we sequence 19 genomes spanning section *Flavi* and compare 31 fungal genomes including 23 *Flavi* species. We reassess their phylogenetic relationships and show that the closest relative of *A. oryzae* is not *A. flavus*, but *A. minisclerotigenes* or *A. aflatoxiformans* and identify high genome diversity, especially in sub-telomeric regions. We predict abundant CAZymes (598 per species) and prolific secondary metabolite gene clusters (73 per species) in section *Flavi*. However, the observed phenotypes (growth characteristics, polysaccharide degradation) do not necessarily correlate with inferences made from the predicted CAZyme content. Our work, including genomic analyses, phenotypic assays, and identification of secondary metabolites, highlights the genetic and metabolic diversity within section *Flavi*.

[1] Department of Biotechnology and Bioengineering, Technical University of Denmark, Søltoft Plads 223, 2800 Kongens Lyngby, Denmark. [2] US Department of Energy Joint Genome Institute, 2800 Mitchell Drive, Walnut Creek, CA 94598, USA. [3] Kikkoman Corporation, 250 Noda, 278-0037 Noda, Japan. [4] Fungal Physiology, Westerdijk Fungal Biodiversity Institute & Fungal Molecular Physiology, Utrecht University, Uppsalalaan 8, 3584 CT Utrecht, The Netherlands. [5] Department of Microbiology, Faculty of Agriculture and Forestry, University of Helsinki, Viikinkaari 9, Helsinki, Finland. [6] US Department of Energy Joint BioEnergy Institute, 5885 Hollis St., Emeryville, CA 94608, USA. [7] Architecture et Fonction des Macromolécules Biologiques, (CNRS UMR 7257, Aix-Marseille University, 163 Avenue de Luminy, Parc Scientifique et Technologique de Luminy, 13288 Marseille, France. [8] Department of Plant and Microbial Biology, University of California, 111 Koshland Hall, Berkeley, CA 94720, USA. [9] Kanazawa Institute of Technology, 3 Chome-1, 924-0838 Yatsukaho, Hakusan-shi, Ishikawa-ken, Japan. [10] Environmental Molecular Sciences Division, Earth and Biological Sciences Directorate, Pacific Northwest National Laboratory, 902 Battelle Blvd, Richland, WA 99354, USA. *email: MRRA@novozymes.com

Aspergillus section *Flavi* encompasses a large number of species, many of which have a significant impact on human life: some species (e.g., *A. oryzae* and *A. sojae*) are routinely used in production of sake, miso, soy sauce, and other fermented foods. Moreover, *A. oryzae* is used industrially for production of enzymes and secondary metabolite production[1–4]. In contrast, other *Flavi* species (e.g., *A. flavus* and *A. parasiticus*) are notorious for producing highly toxic fungal compounds (e.g., aflatoxins), in addition to infecting and damaging crops[5–7]. Furthermore, *A. flavus* has been shown to infect immunocompromised humans, and is currently the second most common cause of human aspergillosis[8,9].

In addition, the section includes less known species that, similar to their (in)famous relatives, display both beneficial and harmful properties. The benefits are found in producers of bioactive compounds (such as the anti-insectant N-alkoxypyridone metabolite, leporin A, from *A. leporis*; an antibiotic with antifungal activity, avenaciolide, from *A. avenaceus*) and enzyme producers (including amylases, proteases, and xylanolytic enzymes in *A. tamarii* and pectin-degrading enzymes in *A. alliaceus*). On the harmful side, plant pathogens (*A. alliaceus* on onion bulb, *A. nomius* on nuts, seeds, and grains) and toxin producers (ochratoxin from *A. alliaceus*, aflatoxin from *A. nomius*) are also found among these less studied *Flavi* species[10,11], for which no genome sequences have previously been available.

Given the importance of section *Flavi*, it is highly valuable to examine the full genetic potential of the section in order to assess alternative species for industrial use, combat pathogenicity, find novel bioactives, and to identify useful enzymes. Prior to this project, whole-genome sequences were only available for five species from section *Flavi* (*A. oryzae*, *A. flavus*, *A. sojae*, *A. luteovirescens* (formerly *A. bombycis*), and *A. parasiticus*[3,12–15]). They all belong to a closely related clade within the section and thus cover only a small part of the diversity.

In this study, as part of the *Aspergillus* genus-sequencing project[16,17], we have generated genome sequences for 18 additional species plus an additional *A. parasiticus* isolate, permitting genomic comparisons across 23 members of section *Flavi* containing at least 29 species[10]. We apply these sequences in tandem with experimental and phenotypic data on secondary metabolite production, growth characteristics, and plant polysaccharide degradation to link phenotypes to genotypes and quantify the genetic potential of the section. Our analysis is useful for (1) exploring novel enzymes and secondary metabolites, (2) optimizing food fermentation and industrial use, and (3) improving food and feed protection and control.

## Results and discussion

**Assessment of 19 newly sequenced section *Flavi* genomes.** In this study, we present the whole-genome sequences of 19 species from *Aspergillus* section *Flavi* (Fig. 1b). Two of these (*A. nomius* and *A. arachidicola*[18,19]) were also published by other groups in parallel to this work. We compare these 19 to previously sequenced section *Flavi* species (*A. oryzae*, *A. flavus*, *A. sojae*, and

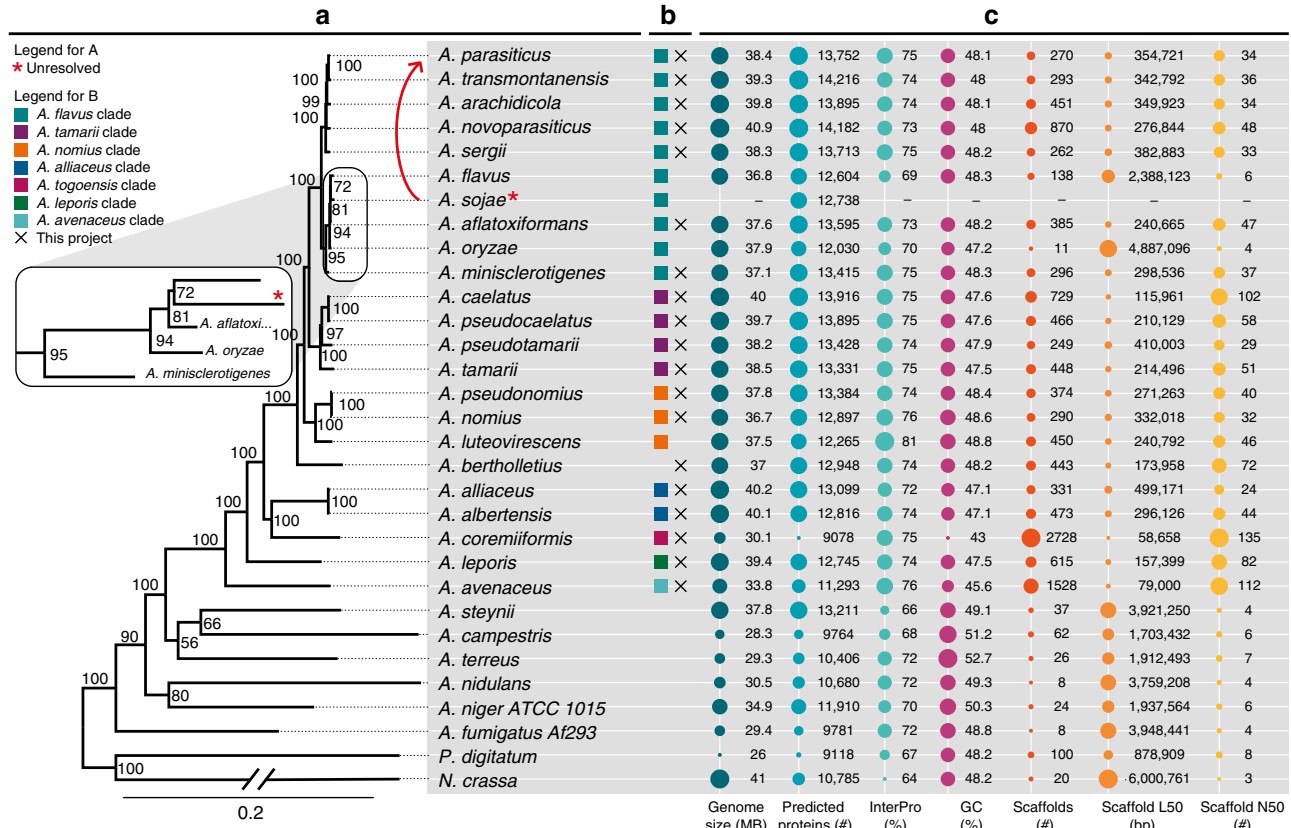

**Fig. 1 Phylogeny and genome statistics of section *Flavi* plus eight other *Aspergillus*, *Penicillium*, and *Neurospora* species. a** Phylogenetic tree constructed using RAxML, MUSCLE, and Gblocks based on 200 monocore genes (a single homolog in each of the species). The red star indicates an uncertain leaf most likely caused by a different gene calling method[98–100], and the arrow shows where *A. sojae* should be placed in the phylogenetic tree. The zoom shows the branching in a clade around *A. oryzae*. **b** The colors illustrate the clades found within section *Flavi* and X indicates species sequenced in this study. Earlier sequenced genomes such as *A. oryzae* and *A. fumigatus* were assembled using optical mapping and genetic maps. **c** Seven bubble plots illustrating key genome numbers and sequencing quality parameter. The bubble sizes have been scaled to each panel and are not comparable across panels.

A. luteovirescens[3,12–14]) as well as eight reference species: six from the rest of genus *Aspergillus* plus *Neurospora crassa* and *Penicillium digitatum* as outgroups (Fig. 1a, b).

As a first basis test, the quality of the genome assemblies was compared based on genome size, GC content, and number of predicted proteins (Fig. 1c). This showed a reasonable draft genome quality with 13 out of the 18 genomes assembled into fewer than 500 scaffolds (Fig. 1c, column 5). One cause of alarm was *A. coremiiformis* with 2728 scaffolds, which made us concerned with the quality of the gene content. However, the genome covers 99.78% of the Benchmarking Universal Single-Copy Orthologs (BUSCO[20]), and 96% of the expressed sequence tag (EST) clusters can be mapped to the genome. We thus conclude that the genome annotation is of a high enough quality for comparisons of the gene content despite the large number of scaffolds.

**Section *Flavi* species generally have expanded genomes**. The genome sizes of *Aspergillus* section *Flavi* are generally large compared with other representative Aspergilli (average of 37.96 Mbp vs. 31.7 Mbp (Fig. 1c)), as was previously reported for *A. oryzae*[21]. One major exception is *A. coremiiformis*, which has both fewer genes and a notably smaller genome, making it unique in the section.

**Multigene phylogeny shows complex heritage of *A. oryzae***. Next we examined the evolutionary relationships in section *Flavi* based on a phylogeny derived from 200 genes (Fig. 1a). The support of the branching within the tree is high (100 out of 100 bootstraps in most branches). The tree confirms that section *Flavi* is a monophyletic group. The clades in Fig. 1a correspond to a previously reported phylogenetic tree based on the beta-tubulin gene[10,11,22] and the distances between sections correspond to previous work[23].

One potential error in the tree is that *A. sojae* is found closest to *A. flavus*, since *A. sojae* is perceived as a domesticated version of *A. parasiticus*. This branching indeed also has the lowest bootstrap value in the tree. The most likely explanation is that since the *A. sojae* gene predictions are based on the *A. flavus* and *A. oryzae* genome annotations[24,25], a bias is created in the predicted genes and this bias is likely reflected in the tree. As a test, we have generated phylogenetic trees using alternative methods not dependent on gene annotation (CVTree[26,27]). These clearly show that *A. sojae* is closest to *A. parasiticus*, both when using whole-genome and proteome sequences (Supplementary Fig. 1 and Supplementary Fig. 2). We hence think that *A. sojae* should be placed next to *A. parasiticus* in the phyogenetic tree as the arrow indicated in Fig. 1a.

Furthermore, *A. oryzae*, perceived as a domesticated version of *A. flavus*[10,28–30], is not directly next to it in the tree. However, it has previously been suggested that *A. oryzae* descends from an ancestor that was the ancestor of *A. minisclerotigenes* or *A. aflatoxiformans*[31]. The phylogeny (Fig. 1a, zoom) supports this suggestion, showing that *A. minisclerotigenes* and *A. aflatoxiformans* are closer relatives of *A. oryzae* than *A. flavus*.

**Analysis of shared proteins confirms high genetic diversity**. In order to examine core features shared by all section *Flavi* species, clades, as well as features of individual species, we made an analysis of shared homologous genes within and across species[16], and sorted these into homologous protein families (Fig. 2). This allowed the identification of (1) The core genome—protein families with at least one member in all compared species. This is expected to cover essential proteins. (2) Section-specific and clade-specific genes—genes that have homologs in all members of a clade/section, but not with any other species. (3) Species-specific genes—genes without homologs in any other species in the comparison.

The *core genome* of all 31 species in this dataset is 2082 protein families. For the 29 *Aspergillus* species this number is 3853, and for section *Flavi* species alone constitutes 4903 protein families. Thus, more than half the genome of the section *Flavi* species varies across the species.

Examining the clade-specific protein families, only very few (27–54) are found (Fig. 2a), which is low compared with section *Nigri* examined previously[16]. As sections *Nigri* and *Flavi* are roughly equally species-rich, this could indicate that the species in section *Flavi* are more distinct. This is supported by the fact that

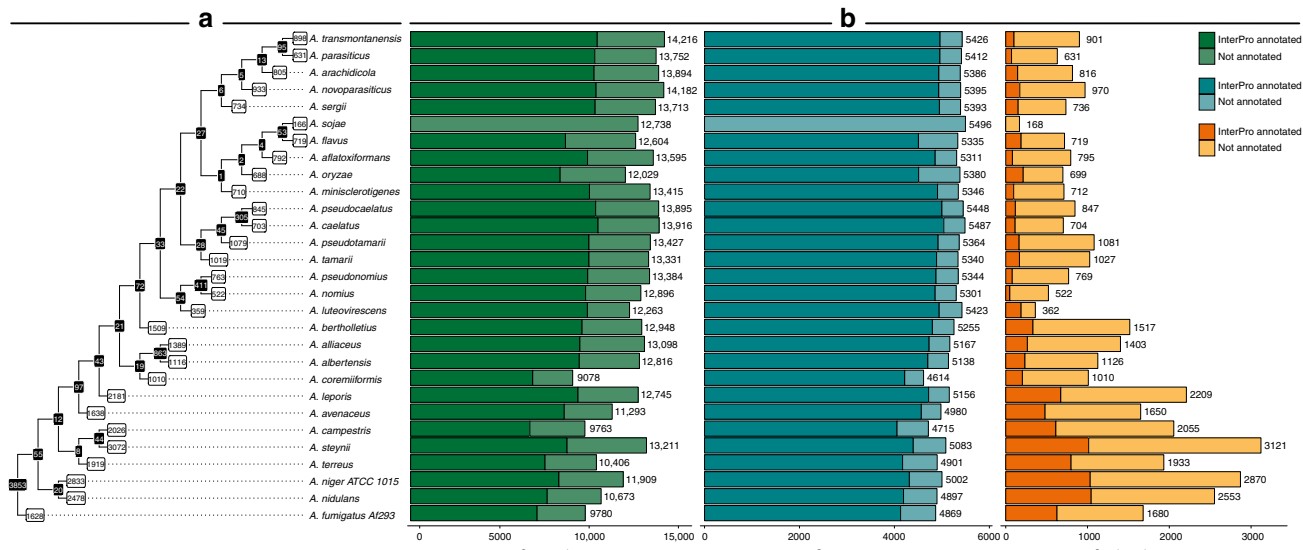

**Fig. 2 Core-specific, section-specific, and clade-specific and species-unique genes. a** A dendrogram representing the phylogenetic relationship between the 29 Aspergilli. The black boxes in the nodes represent the homologous protein families shared among the species branching from that node. The white boxes at the tips represent the protein families unique to that individual species. **b** A barplot showing the number of total (green), core (turquoise), and species-specific (orange) proteins for each species. The dark shading illustrates the number of proteins with a least one functional annotation based on InterPro[32].

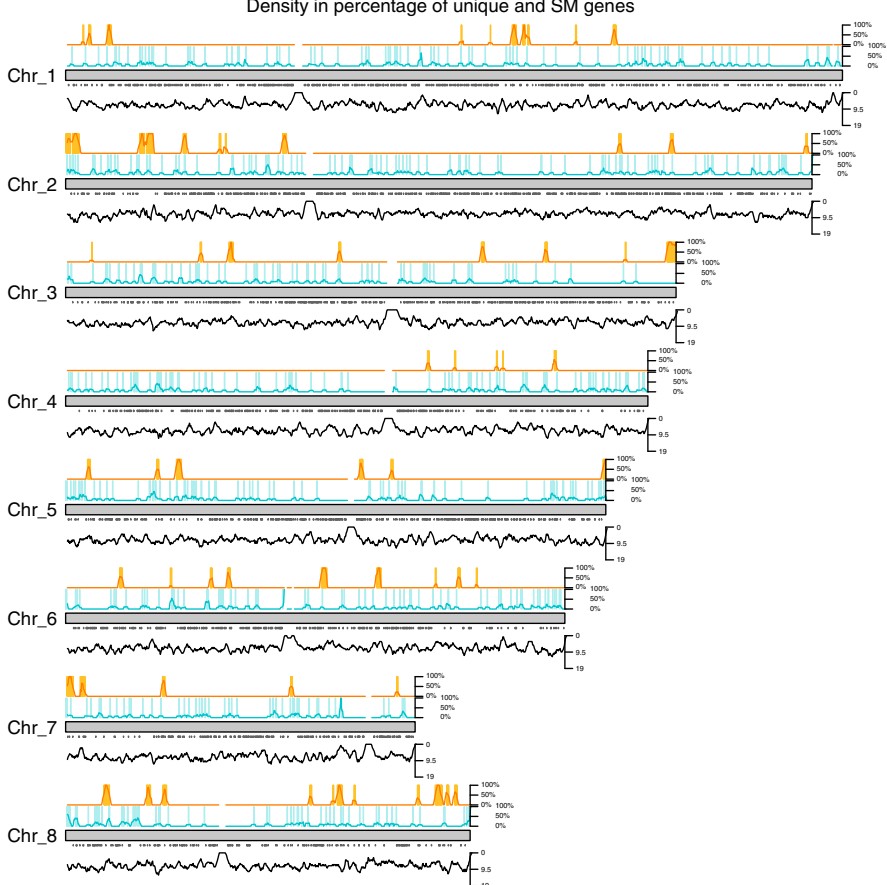

**Fig. 3 Location of species unique and secondary metabolite genes in the *A. oryzae* genome.** The gray bars represent the *A. oryzae* genome. Above the chromosome the species-specific (turquoise) and secondary metabolite genes (orange) are mapped to the genome, each line represents a gene. The curve shows the percentage of the density calculated from the total number of genes within 30 kbp in steps of 5 kb. Below the genome, the core genes are mapped by the gray dots and the density of the total number of genes is shown by the black graph (with a window of 30 kbp).

the number of species-specific genes are very high (166–2181), where we see 166 (*A. sojae*) to be an artificially low number, due to the gene calling in this genome being based on *A. flavus* and *A. oryzae* genomes.

**Species-specific genes often encode regulation and P450s.** We wanted to see whether the species-specific genes could be linked to known *Flavi* functions such as food fermentation and plant and human pathogenicity. In order to do this, we examined predicted functions of the species-specific genes using InterPro, GO and KOG annotations[32–35]. The portion with a functional annotation was low; 20, 12, and 9% for InterPro, GO, and KOG, respectively; in total 21% had an annotation (Supplementary Figs. 3–5). This is a very high—but not unusual—percentage of unidentifiable functions.

We will focus on InterPro since it covers more genes: the most common InterPro functions include transcription factors, protein kinases, transporters, and P450s (Supplementary Fig. 3), which are also significantly overrepresented. While these traits cannot directly be linked to food fermentation and pathogenicity, regulation is involved in adaptation and P450s play roles in both substrate degradation and production of bioactive compounds, both of which are relevant for fungal pathogenicity.

**Species genes are over-represented in sub-telomeric regions.** It has been shown that the sub-telomeric sequences are extensively rearranged regions in *A. nidulans, A. oryzae,* and *A. fumigatus*[21].

This is also seen in mammals, nematodes, and yeasts[36]. Previous studies[37,38] showed that sub-telomeric regions have a bias for unique, diverged, or missing genes. Another study has shown secondary metabolite gene clusters (SMGCs) to be enriched in sub-telomeric regions in *A. nidulans* and *A. fumigatus*[21].

We therefore examined the gene density and location of species-specific genes, secondary metabolite clusters, and core genome, by using the telomere-to-telomere *A. oryzae* genome as a reference in order to assess the potential overrepresentation of these genes in the sub-telomeric regions (Fig. 3).

Both visual inspection and Fisher's exact test confirmed that both species-specific ($p$-value = 7.266e–07) and SMGCs ($p$-value < 2.2e–16) are enriched toward the sub-telomeric regions (100 kbp from the chromosomal ends), where core genes are found less often at the sub-telomeric regions. The fact that the species-specific genes are not randomly distributed argues against that they are simply annotation or gene modeling errors, therefore indicating that they are, indeed, legitimate genes. The distribution of the species-specific genes suggests that new genes are more frequently successfully incorporated into the sub-telomeric regions than other locations. Whether this is the result of a selection for the sub-telomeric region, or a counterselection against other regions, or both, the data do not reveal.

**Synteny analysis reveales islands of highly variable gene content.** Syntenic and non-syntenic regions are another factor to consider when analyzing genome location. It has been shown that the *A. oryzae* genome has a mosaic pattern of syntenic and

**Table 1 Percentage of genome with conserved synteny relative to *A. oryzae*.**

| Species | # Syntenic genes | % of *A. oryzae* |
| --- | --- | --- |
| *A. parasiticus* | 8199 | 68.15 |
| *A. transmontanensis* | 8238 | 68.48 |
| *A. arachidicola* | 8817 | 73.29 |
| *A. novoparasiticus* | 8102 | 67.35 |
| *A. sergii* | 8091 | 67.26 |
| *A. flavus* | 8686 | 72.20 |
| *A. aflatoxiformans* | 9094 | 75.59 |
| *A. oryzae* | – | – |
| *A. minisclerotigenes* | 8498 | 70.64 |
| *A. caelatus* | 7411 | 61.60 |
| *A. pseudocaelatus* | 7503 | 62.37 |
| *A. pseudotamarii* | 7494 | 62.29 |
| *A. tamarii* | 7471 | 62.10 |
| *A. pseudonomius* | 7179 | 59.68 |
| *A. nomius* | 7269 | 60.42 |
| *A. luteovirescens* | 7863 | 65.36 |
| *A. bertholletius* | 6801 | 56.53 |
| *A. alliaceus* | 6021 | 50.05 |
| *A. albertensis* | 5998 | 49.86 |
| *A. coremiiformis* | 5425 | 45.10 |
| *A. leporis* | 5800 | 48.21 |
| *A. avenaceus* | 5351 | 44.48 |
| *A. nidulans* | 4272 | 35.51 |
| *A. fumigatus* | 4876 | 40.53 |

non-syntenic regions relative to distantly related Aspergilli[1,2]. We examined the synteny across section *Flavi* and into *A. nidulans* and *A. fumigatus* using *A. oryzae* RIB40 as reference (Table 1). This analysis supports our earlier finding that *A. oryzae* is closely related to *A. aflatoxiformans* than *A. flavus*.

An overview of shared syntenic genes are illustrated in Supplementary Fig. 6. In general, there are fewer regions of synteny toward the telomeric ends as previously seen[1,2] in a comparison of *A. nidulans*, *A. fumigatus*, and *A. oryzae*. We further observed that chromosomes 1 and 2 have a very high degree of conserved synteny, while chromosomes 6 and 8 have a much lower conservation of synteny.

We find dense islands of non-syntenic genes in non-sub-telomeric regions on chromosomes 4, 6, and 8. These could be caused by horizontal gene transfer (HGT), gene shuffling, or de novo gene formation. We investigated for HGTs using BLASTp to examine the best hits in the NCBI nonredundant database. Recent HGTs are expected to have high sequence identity with another group of species where it would have been transferred from, and not be found in the closely related species[39]. None of these islands showed signs of recent HGTs. Furthermore, only 23 of the 80 genes in the non-syntenic blocks were *A. oryzae*-specific. It thus seems likely that these non-syntenic islands are caused by a mix of significant rearrangements, duplication events, and the emergence of *A. oryzae*-specific genes.

Taken together, the fact that we observe some very conserved chromosomes and some highly rearranged non-syntenic blocks could indicate an evolutionary pressure for stability in some regions while other regions are frequently subject to gene shuffling and rearrangements, i.e., rearrangement hot spots.

**Section *Flavi* is a rich source of carbohydrate-active enzymes**. Carbohydrate-Active enZymes (CAZymes) are essential for what carbon sources a species can degrade and utilize. Within section *Flavi* the CAZymes/carbon utilization is mainly described for *A. oryzae*[1,2,40] and to a lesser extent for *A. flavus*[41–45] and *A.*

*sojae*[46,47], while only incidental studies have been performed with other species of this group[48–54], often describing production or characterization of a certain CAZyme activity or protein, respectively.

We used the CAZy database to predict the CAZyme content in the genomes of the section (Fig. 4). A total of 13,759 CAZymes were predicted for the 23 *Flavi* species (average 598/species). This is quite rich compared with included reference Aspergilli (508/species).

It is clear from this analysis that there is a distinct difference between the clades of section *Flavi* (Fig. 4b), showing again a variation in gene content in the section.

**Variable CAZyme content does not reflect the ability to degrade plant biomass**. To evaluate the actual carbon utilization ability across section *Flavi*, we performed growth profiling of 31 species (29 Aspergilli, including 23 species from section *Flavi*) on 35 plant biomass-related substrates (Fig. 5, Supplementary Data 1) and compared this with the CAZyme gene content prediction related to plant biomass degradation (Supplementary Data 2). In a previous study, the variation in growth between distantly related Aspergilli could be linked to differences in CAZyme gene content[55], but this was not the case for closer related species from *Aspergillus* section *Nigri*[16].

Glucose resulted in the best growth of all monosaccharides for all species and was therefore used as an internal reference for growth (Supplementary Fig. 7). Growth on other carbon sources was compared with growth on D-glucose and this relative difference was compared between the species. Growth on monosaccharides was largely similar between the species of section *Flavi* (Fig. 5, Supplementary Fig. 7, and Supplementary Data 1).

The CAZyme sets related to plant biomass degradation are overall highly similar for section *Flavi* (Fig. 5), with the exception of *A. coremiiformis*, which has a strongly reduced gene set. This is mainly due to reduction in glycoside hydrolase families, but also a number of families related to pectin, xylan, and xyloglucan degradation. Surprisingly, this species showed better relative growth on xylan than most other species, while the growth on other polysaccharides was mainly similar to that of section *Flavi*. Thus, the reduced gene set has not reduced its ability to degrade plant biomass. This could be similar to the case of *T. reesei*, which also has a reduced CAZyme gene set, but produces the corresponding enzymes at very high levels[56]. However, the origin of this approach is likely very different as its CAZyme content was shaped by loss and then massive HGT gain of plant cell wall degrading enzymes[57], while no indications for this are present for *A. coremiiformis*.

Hydrolytic differences are clade-specific within section *Flavi* (Supplementary Data 2). The *A. togoensis* clade has a reduced set of xylanolytic and xyloglucanolytic genes, but this is not reflected in the growth. In contrast, GH115 (alpha-glucuronidase) genes are expanded in clades *A. flavus*, *A. tamarii*, and *A. nomius* (xylanolytic enzymes or activity have been reported from several species from these clades[49–51,53,58–62]), GH62 (arabinoxylan arabinofuranohydrolase) was expanded in clade *A. leporis*, and clades *A. leporis* and *A. avenaceus* were the only clades with CE15 (glucuronoyl esterases), which were also found in *Aspergillus* species outside section *Flavi*.

The galactomannan degrading ability was nearly fully conserved in section *Flavi*, but interestingly growth on guar gum that consists mainly of galactomannan was variable between the species. Similarly, the reduced amylolytic ability of clades *A. togoensis* and *A. avenaceus* did not result in reduced growth on starch or maltose.

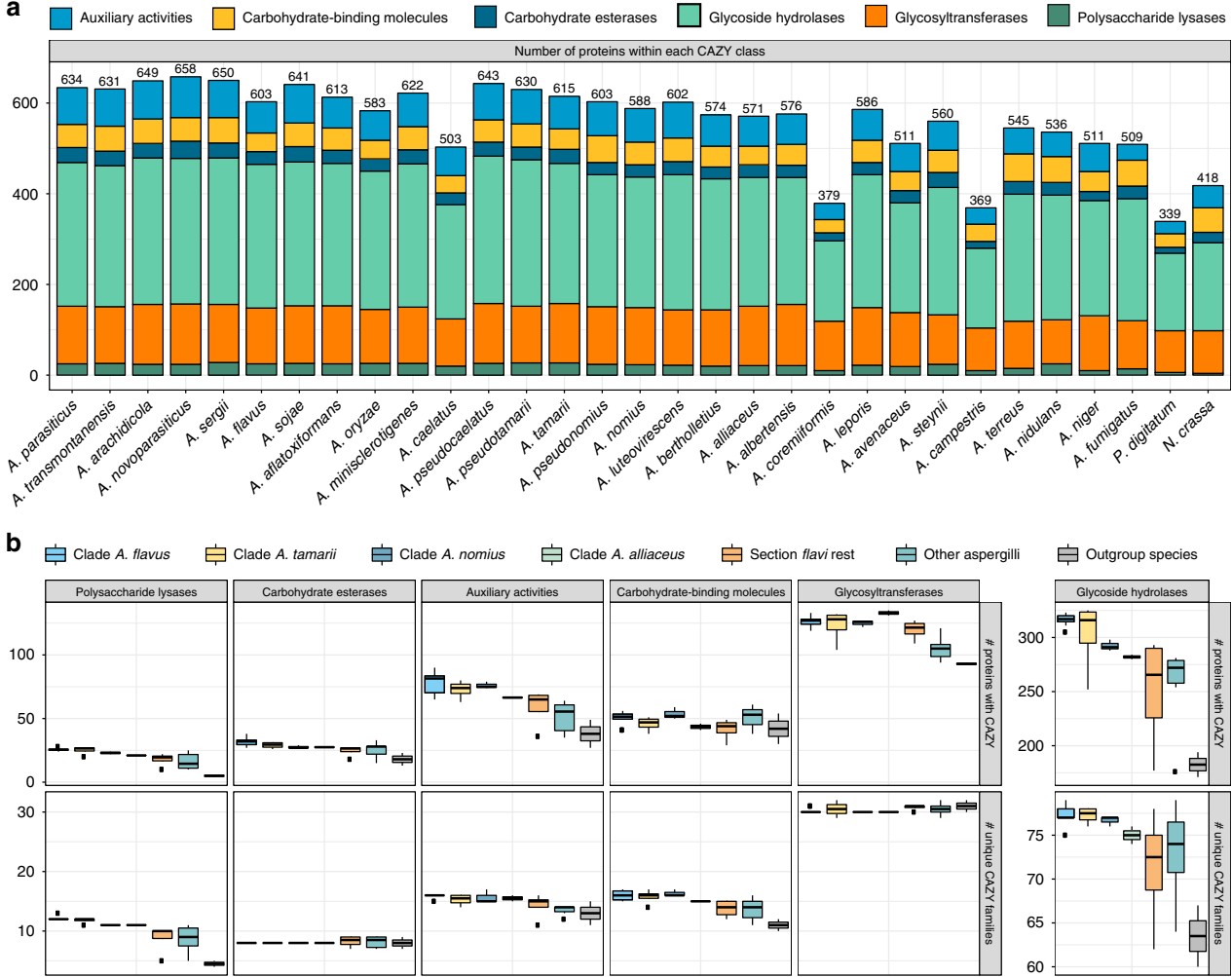

**Fig. 4 Carbohydrate-active enzymes (CAZymes) in section *Flavi*. a** The total number of CAZymes in each species distributed on six categories of enzyme activity: auxiliary activities, carbohydrate-binding molecules, carbohydrate esterases, glycoside hydrolases, glycosyltransferases, and polysaccharide lyases. **b** Boxplot representing the diversity of CAZyme family content and abundance among clade *A. flavus* (light blue), *A. tamarii* (yellow), *A. nomius* (dark blue), *A. alliaceus* (light turquoise), the rest of the *Flavi* section (orange), other Aspergilli (dark turquoise), and non-Aspergillus species (gray). For each CAZyme class the total number of CAZymes (top row) and the number of unique CAZyme families (bottom row) are displayed. In the boxplot the midline represents the median, the upper and lower limit of the box represents the third and first quartile, and the whiskers extend up to 1.5 times the interquartile.

Variation was observed in the number of pectinolytic genes. The most pronounced differences were the absence of PL11 (rhamnogalacturonan lyase) genes from most species of section *Flavi*, and the expansion of GH78 (alpha-rhamnosidase) in clades *A. flavus* and *A. tamarii*. However, these differences and the smaller ones in other families did not result in large variation in growth on pectin.

More obvious differences were present during growth on cellobiose, lactose, and lignin. Most species grew poorly on cellobiose despite similar numbers of beta-glucosidase-encoding genes in most species (Supplementary Data 2). Similarly, only *A. arachidicola*, and to a lesser extent *A. albertensis* grew well on lactose, while the number of beta-galactosidases in these species is similar to that of the other species. Most interesting was the finding that *A. albertensis* grew as well on lignin as on D-glucose, suggesting potential applications in biofuel production.

In summary, CAZyme potential in section *Flavi* is largely conserved (with the exception of *A. coremiiformis*) with some variations in copy numbers, but the genomic potential and variations are not necessarily reflected in the growth. It is therefore likely that as suggested previously[55], the observed differences are largely at the regulatory level.

**CAZyme family GH28 is inflated in clade *A. flavus*.** We were particularly interested in GH28 CAZymes, as they are important for food fermentation and the quality of the final fermented product[63]. A phylogenetic tree was created of all members of GH28 from section *Flavi* (Supplementary Fig. 8). The tree consists of 429 proteins, on average 18.7 per species.

Within the tree there are different groupings. Five groups have members from all 23 species, nine groups are missing one to four species (usually *A. coremiiformis* and *A. caelatus*), and two groups are specific to the *A. flavus*, *A. tamarii*, and *A. nomius* clades. Last there are eight groups containing 2–13 species, which do not follow the phylogeny—suggesting these to be sources of GH28 variation.

In general, species from clade *A. flavus* have a high number of GH28 members. *A. sojae* is known to have a high number of GH28, which is also seen here with 24 members; however, *A. sergii* has an even higher number with 25 members. It could be interesting to investigate if this could be exploited either by using *A. sergii* as a new species in food fermentation and/or as a source of novel enzymes.

**Analysis of secondary metabolism.** The genus *Aspergillus* is known to produce a large number of SMs and the number of

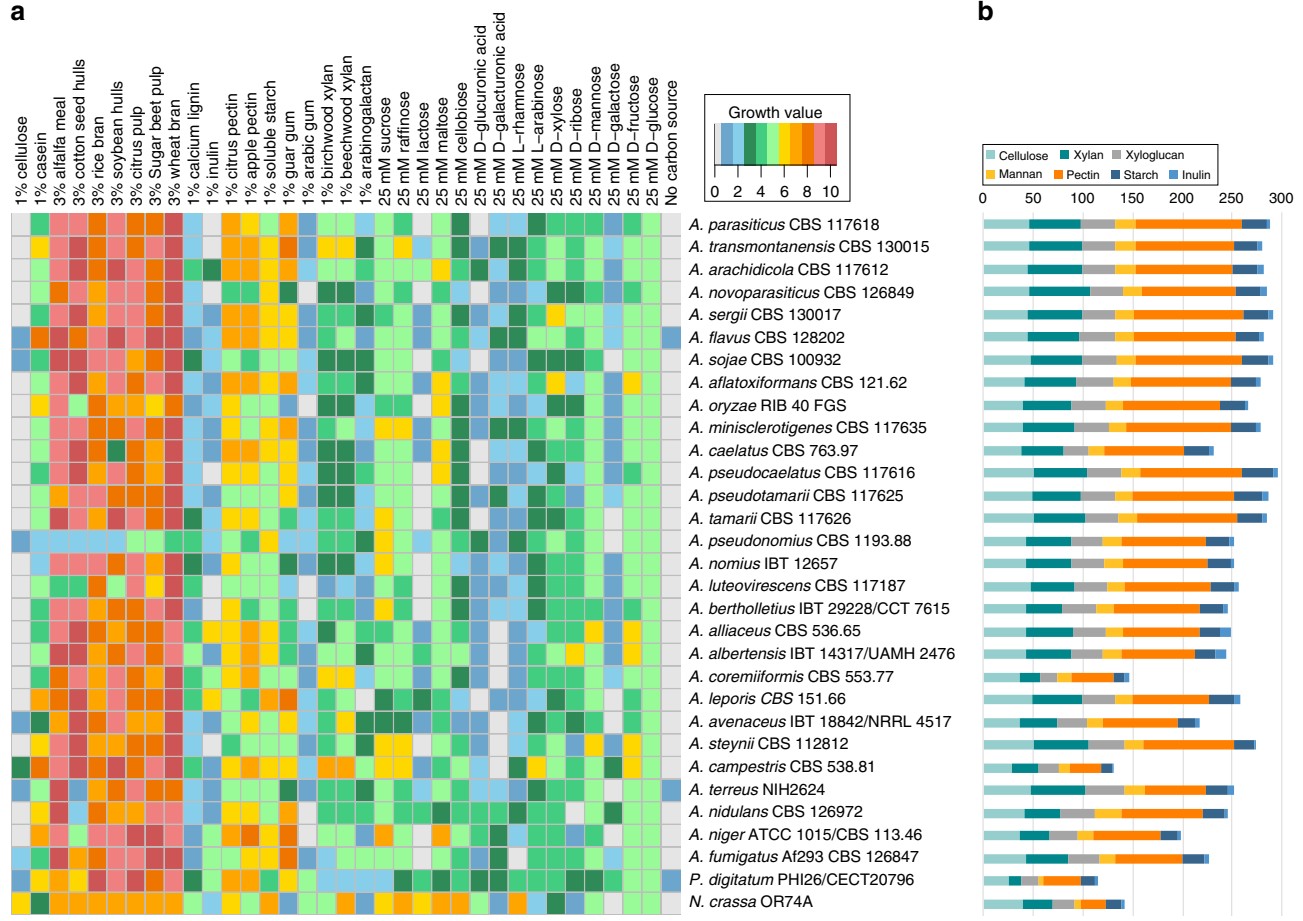

**Fig. 5 Carbohydrate-active enzymes in section *Flavi* sorted according to the phylogram of Fig. 1. a** Heatmap representing the growth profiles of 23 *Flavi* species and 8 additional species on 35 different media. **b** Comparison of the CAZyme sets related to plant biomass degradation in the genomes of species from *Aspergillus* section *Flavi*, and some other fungi. The colors reflect the polysaccharides the enzymes are active toward.

predicted SMGCs is even higher. The majority of predicted SMGCs are uncharacterized and therefore have the potential to produce a diversity of novel, bioactive compounds. We examined the diversity and potential for SM production in section *Flavi*, both quantitatively in terms of numbers of clusters, and qualitatively in terms of the compounds these clusters could potentially produce.

**Secondary metabolism in section *Flavi* is diverse and prolific.** To quantitatively assess the potential for SM production, SMGCs were predicted using a SMURF-like prediction tool[64] for all species except *N. crassa* and *A. sojae*, since these were sequenced by other methods and with dissimilar gene calling methods (Fig. 6c). Within the 28 *Aspergillus* species, there is a total of 1972 predicted SMGCs and for section *Flavi* genomes, the total is 1606 SMGCs (73/species). This is more than 15 extra per species compared with the very prolific *Penicillium* genus[65].

We wanted to examine how unique the SMGCs are, and thus constructed families of SMGCs (Supplementary Data 3). For the entire dataset, we could collapse it into 477 SMGC families, and for section *Flavi* 308 SMGC families. Out of these, 150 SMGC clusters are only found in one section *Flavi* species (Fig. 6a), showing a large number of unique clusters in each species (6.8 unique SMGCs/species). Compared with *Aspergillus* section *Nigri*, the number of clusters per species in this study is slightly lower, but the number of members in each SMGC family is also lower, demonstrating greater diversity in secondary metabolism in section *Flavi* compared with section *Nigri*.

**Dereplicating secondary metabolism predicts toxin producers.** To assess the potential for SM production qualitatively, we used a pipeline of "genetic dereplication" where predicted clusters are associated with verified characterized clusters (from the MIBiG database[66]) in a guilt-by-association method[67]. Based on this, 20 cluster families were coupled to a compound family (Fig. 6b). Some cluster families were found in all or nearly all *Flavi* genomes, e.g., those similar to the naphthopyrone[68], nidulanin A[69], azanigerone[70], 4,4′-piperazine-2,5-diyldimethyl-bis-phenol, and aflavarin[71]/endocrocin[72,73] clusters. Most families generally follow the phylogenetic groups, suggesting a loss-based distribution pattern, but some, like the SMGC families similar to the asperfuranone[74], pseurotin A[75], or fumagillin[76] clusters did not follow the phylogeny. Moreover, potential producers of known toxins such as aflatoxin and aspirochlorine were identified (Fig. 6b).

**Combination of data and analysis links a compound to a cluster.** Extending from the known SMGC clusters, we were interested in linking compounds and clusters based on the presence/absence pattern of produced compounds and predicted clusters. We therefore created a heatmap of all the cluster families found in at least five species, added the predicted compound families from the MIBiG dereplication, in addition to manually curated compound families from a literature survey (Supplementary Fig. 9). In addition to this, we measured the SM production of the *Flavi* species (Supplementary Data 4).

Of particular interest was miyakamides. They are originally isolated from an *A. flavus* isolate and shown to have antibiotic

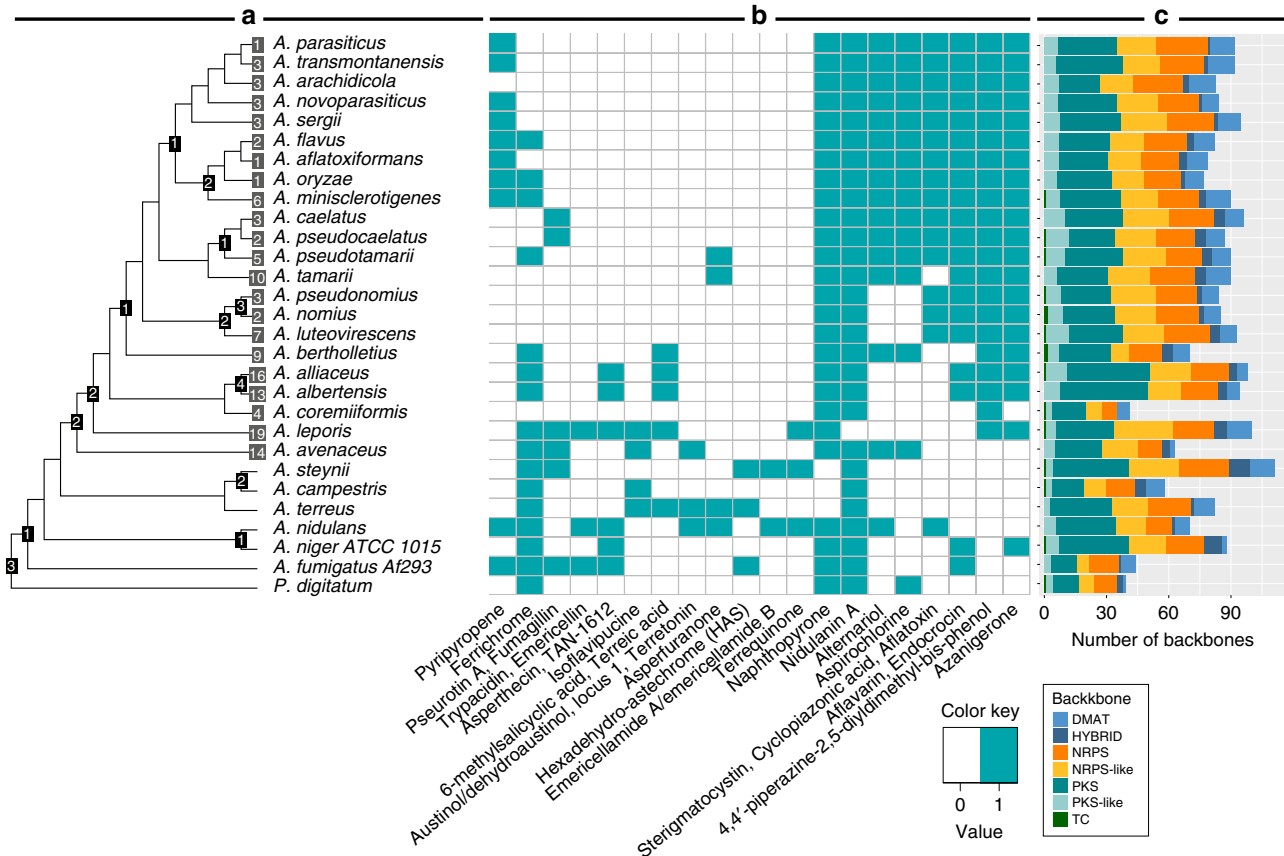

**Fig. 6 Dereplication of known compounds and predicted secondary metabolite backbone genes per species. a** A dendrogram representing the phylogenetic relationship between the species. The black boxes in the nodes represent the secondary metabolite gene cluster (SMGC) families shared among the species branching from that node. If there is no black box there are zero clusters shared. The gray boxes at the tips show the number of unique SMGC families only found in one species for the *Flavi* section. **b** Matrix indicating the presence and absence of SMGC families coupled to known clusters from the MIBiG database[66] for each species. Overview of the cluster family for aflatoxin can be found in Supplementary Figure 11. **c** Predicted secondary metabolite genes for each species divided by the backbone enzyme. DMAT: dimethylallyltransferase (prenyl transferases), HYBRID: a backbone gene containing domains from NRPS and PKS backbones, NRPS: non-ribosomal peptide synthetase, NRPS-like: non-ribosomal peptide synthetase like, containing at least two NRPS-specific domains and another domain or one NRPS A domain in combination with NAD binding 4 domain or short-chain dehydrogenase, PKS: polyketide synthase, PKS-like: polyketide synthase like, containing at least two PKS-specific domains and another domain, TC: terpene cyclase.

properties[77], but the biosynthetic gene cluster is not known. Our chemical analysis showed production in *A. sojae, A. nomius, A. parasiticus, A. novoparasiticus*, and *A. transmontanensis*.

We performed retro-biosynthesis from the chemical structure and predicted that the biosynthetic gene cluster should contain a nonribosomal peptide synthetase (NRPS) with 2–3 adenylation domains (since two of the three amino acids are similar), an N-methyltransferase, an acetyltransferase, and potentially a decarboxylase/dehydrogenase (Supplementary Fig. 10A). Searching for cluster families with members in all the miyakamide-producing species having NRPS backbones with 2–3 adenylation domains and a methyltransferase domain, only one cluster family met the requirements. The cluster family has a NRPS backbone with a methyltransferase domain, three A domains in most species, and two in *A. novoparasiticus*. The prediction of only two A domains is most likely caused by annotation error since the sequence similarity is conserved before the start of the gene (Supplementary Fig. 10B). The size of the predicted cluster is 1–9 genes, the difference is likely caused by SMGC prediction errors (Synteny plot in Supplementary Fig. 10B). The synteny plot shows that the NRPS and two small genes with unknown function are widely conserved. We thus propose that the identified NRPS along with the two conserved genes of unknown function are likely candidates for miyakamide biosynthesis.

**The aflatoxin biosynthetic gene cluster is highly conserved.** Perhaps the best known secondary metabolite in section *Flavi* is the highly carcinogenic aflatoxin. Aflatoxins are known to be produced by many section *Flavi* species (*A. arachidicola, A. luteovirescens, A. flavus, A. minisclerotigenes, A. nomius, A. aflatoxiformans, A. pseudocaelatus, A. pseudonomius, A. pseudotamarii*, and some *A. oryzae* isolates)[4,10].

The dereplication analysis (Fig. 6b) identified a SMGC family predicted to be involved in sterigmatocystin and aflatoxin production, which is all the species in the *A. flavus, A. nomius,* and *A. tamarii* clades except *A. tamarii*. A synteny plot of the SMGC family (Supplementary Fig. 11) shows that the cluster is extremely well conserved with no rearrangements and a high alignment identity for the aflatoxin genes. Only *A. caelatus* has a truncated form with only the *aflB, aflC,* and *aflD* genes and *A. tamarii* seems to have a complete loss of the cluster. Interestingly, most of the predicted clusters did not include the *aflP* and *aflQ* genes that are responsible for the last step of aflatoxin biosynthesis. We searched the genomes for *aflP* (Supplementary Fig. 12), and found it in all genomes, but with different start sites and extra sequence in the middle of the proteins. RNA-seq data support these models (Supplementary Fig. 13) and suggest errors in the *A. flavus* gene models. Similarly, the *aflQ* gene is found in all the other species, but 5–10 genes away from the predicted

clusters. Thus, detailed analysis shows that all these species have the genes required for aflatoxin biosynthesis.

## Conclusion

We de novo sequenced the genomes of species representing various parts of the *Flavi* section, which allowed a section-wide comparison illustrating the similarities and diversity within the section. We show that *A. oryzae* is closely related to *A. minisclerotigenes* or *A. aflatoxiformans* based on a 200-gene phylogeny.

Members of the *Flavi* section have a large genome size compared with other Aspergilli. The large genome is reflected in the high number of SMGCs and CAZymes that could be a source of novel compounds and enzymes in the future.

We have shown that the aflatoxin cluster is highly conserved both concerning identity and synteny in the *A. flavus*, *A. nomius* clade, and partly in the *A. tamarii* clade where the cluster is partly lost in *A. caelatus* and completely lost in *A. tamarii*.

The number of species unique proteins is varying, but even with the very closely related *A. flavus* clade, most species have above 700 unique proteins illustrating the high diversity. Localization analysis of *A. oryzae* has shown the distribution of species unique genes and SMGC across the genome but with a higher density in the sub-telomeric ends. Synteny analyses have highlighted some tendencies of some highly conserved chromosomes and a few dense non-syntenic blocks that could represent rearrangement hot spots.

Overall the data and analysis presented here provide the fungal research community with a substantial resource, and set the stage for future research in the field.

## Methods

**Fungal strains**. The species examined in this study (Supplementary Table 1) were from the IBT Culture Collection of Fungi at the Technical University of Denmark (DTU) or from the Westerdijk Fungal Biodiversity Institute (CBS), unless otherwise noted. Strains can be obtained from these sources.

**Purification of DNA and RNA**. For all sequences generated for this study (Supplementary Table 1), spores were defrosted from storage at −80 °C and inoculated onto solid CYA medium. Fresh spores were harvested after 7–10 days and suspended in a 0.1% Tween solution. Spores were stored in solution at 5 °C for up to 3 weeks. Biomass for all fungal strains was obtained from shake flasks containing 200 ml of complex medium, CYA, MEAox, or CY20 depending on the strain (see Supplementary Table 1) cultivated for 5–10 days at 30 °C. Biomass was isolated by filtering through Miracloth (Millipore, 475855-1R), freeze dried, and stored at 80 °C. DNA isolation was performed using a modified version of the standard phenol extraction (see ref. [78] and below) and checked for quality and concentration using a NanoDrop (BioNordika). RNA isolation was performed using the Qiagen RNeasy Plant Mini Kit according to the manufacturer's instructions. A sample of frozen biomass was subsequently used for RNA purification. First, hyphae were transferred to a 2 ml microtube together with a 5-mm steel bead (Qiagen), placed in liquid nitrogen, then lysed using the Qiagen TissueLyser LT at 45 Hz for 50 s. Then the Qiagen RNeasy Mini Plus Kit was used to isolate RNA. RLT Plus buffer (with 2-mercaptoethanol) was added to the samples, vortexed, and spun down. The lysate was then used in step 4 in the instructions provided by the manufacturer, and the protocol was followed from this step. For genomic DNA, a protocol based on Fulton et al.[79] was used. The same procedure was used previously[16,80].

**DNA and RNA sequencing and assembly**. All genomes and transcriptomes in this study were sequenced with Illumina. For all genomic Illumina libraries, 100 ng of DNA was sheared to 270-bp fragments using the Covaris LE220 (Covaris) and size selected using SPRI beads (Beckman Coulter). The fragments were treated with end repair and A tailing and ligated to Illumina-compatible adapters (IDT) using the KAPA-Illumina library creation kit (KAPA Biosystems).

For transcriptomes, stranded complementary DNA libraries were generated using the Illumina TruSeq Stranded Total RNA LT Sample Prep Kit. Messenger RNA (mRNA) was purified from 1 µg of total RNA using magnetic beads containing poly(T) oligos. mRNA was fragmented using divalent cations and high temperature. The fragmented RNA was reverse transcribed using random hexamers and SSII (Invitrogen) followed by second-strand synthesis. The fragmented complementary DNA was treated with end repair, A tailing, adapter ligation, and ten cycles of PCR.

The prepared libraries were quantified using KAPA Biosystems' next-generation sequencing library quantitative PCR kit and run on a Roche LightCycler 480 real-time PCR instrument. The quantified libraries were then multiplexed with other libraries, and library pools were prepared for sequencing on the Illumina HiSeq sequencing platform using a TruSeq paired-end cluster kit, v3, and Illumina's cBot instrument to generate clustered flow cells for sequencing. Sequencing of the flow cells was performed on the Illumina HiSeq2000 sequencer using a TruSeq SBS sequencing kit, v3, following a 2 × 150 indexed run recipe.

After sequencing, the genomic FASTQ files were quality control-filtered to remove artifacts/process contamination and assembled using Velvet54. The resulting assemblies were used to create in silico long mate-pair libraries with inserts of 3000 ± 90 bp, which were then assembled with the target FASTQ using AllPathsLG release version R4771055. Illumina transcriptome reads were assembled into consensus sequences using Rnnotator v3.3.256.

**Genome annotation**. All genomes were annotated using the JGI annotation pipeline[81,82] as previously described[16,80]. Genome assembly and annotations are available at the JGI fungal genome portal MycoCosm[81] (see URLs) and have been deposited in the DNA Data Bank of Japan (DDBJ)/European Molecular Biology Laboratory (EMBL)/GenBank under the accession numbers provided in the Data Availability Statement.

**Homologous protein families**. All predicted proteins from the 31 genomes used in this study were aligned using the BLASTp function from the BLAST + suite version 2.2.27 with an (e-value < $10^{10}$). The resulting 961 whole-genome BLAST tables were analyzed to identify homologous proteins and group them into families as described previously[16].

Protein families containing at least one protein from all species were defined as core families, while species-unique families were defined as families containing one or more protein(s) from only one species.

**Functional annotation**. Functional domains were identified in all the proteins using InterPro[32], GO[34], and KOG[35].

**Phylogeny**. Monocore genes were identified as protein families having exactly one member in each species. Each protein family was aligned using MUSCLE version 3.8.31 (default settings) and then trimmed using gblocks version 0.91b ($-t = p$ $-b4 = 5$ $-b5 =$ h). Following 200 of these monocore sequences (with length between 150 and 1000 AA) were selected randomly and concatenated and used to construct a phylogentic tree using RaxML version 8.2.8 using the PROT-GAMMAWAG substitution model and 1000 bootstraps.

**Prediction of CAZymes**. CAZymes were predicted using the CAZymes database (CAZy, www.cazy.org[83]) and the method described in our previous work[16]. Each *Aspergillus* protein model was compared using BLASTp with proteins listed in the CAZYmes database (CAZy)[83,84]. Models with over 50% identity over the entire length of an entry in CAZy were directly assigned to the same family (or subfamily when relevant). Proteins with less than 50% identity to a protein in CAZy were all manually inspected, and conserved features, such as the catalytic residues, were searched whenever known. Because 30% sequence identity results in widely different e-values (from nonsignificant to highly significant), for CAZy family assignments, we examined sequence conservation (percentage identity over CAZy domain length). Sequence alignments with isolated functional domains were performed in the case of multimodular CAZymes. The same methods were used for *Penicillium digitatum* and *Neurospora crassa*.

**Prediction of secondary metabolite gene clusters**. Secondary metabolite gene clusters (SMGCs) and SMGC families were predicted based on the SMURF algorithm[64] and the method described in our previous work[16]. For the prediction of SMGCs, we developed a command-line Python script roughly following the SMURF algorithm:

According to SMURF the following genes were predicted as "backbone" genes:

Genes that have at least three PFAM domains—ketoacyl-synt (PF00109), Ketoacyl-synt_C (PF02801), and Acyl_transf_1 (PF00698)—were predicted as 'PKS' genes.

Genes that have ketoacyl-synt (PF00109) and Ketoacyl-synt_C (PF02801) but not Acyl_transf_1 (PF00698) were predicted as "PKS-like" genes.

Genes that have at least the three domains AMP-binding (PF00501), PP-binding (PF00550), and Condensation (PF00668) were predicted as "NRPS" genes.

Genes that have an AMP-binding (PF00501) domain and at least one of the domains PP-binding (PF00550), Condensation (PF00668), NAD_binding_4 (PF07993), and Epimerase (PF01370) were predicted as "NRPS-like" genes.

Genes that have both "PKS" and "NRPS" domains were predicted as "Hybrid" genes.

Genes that have a Trp_DMAT domain were predicted as "DMAT" genes.

Genes that have Terpene_synth (PF01397) or Terpene_synth_C (PF03936) domains were predicted as "Terpene cyclase/synthase" genes.

Secondary metabolite-specific PFAM domains were taken from Supplementary Data 1 of the SMURF paper[64]. As input, the program takes genomic coordinates and the annotated PFAM domains of the predicted genes. Based on the multidomain PFAM composition of identified "backbone" genes, it can predict seven types of secondary metabolite clusters: (1) polyketide synthases (PKSs), (2) PKS-like, (3) non-ribosomal peptide synthetases (NRPSs), (4) NRPS-like, (5) hybrid PKS–NRPS, (6) prenyltransferases (DMATS), and (7) terpene cyclases (TCs). Besides backbone genes, PFAM domains, which are enriched in experimentally identified secondary metabolite clusters (secondary metabolite-specific PFAMs), were used in determining the borders of gene clusters. The maximum allowed size of intergenic regions in a cluster was set to 3 kb, and each predicted cluster was allowed to have up to six genes without secondary metabolite-specific domains.

SMGC families were created based on the SMURF prediction comparisons using BLASTp (BLAST + suite version 2.2.27, e-value ≤ 1 × 10⁻¹⁰). Subsequently, a score based on BLASTp identity and shared proteins was created to determine the similarity between gene clusters as depicted in the formula below. Using these scores, we created a weighted network of SMGC clusters and used a random walk community detection algorithm (R version 3.3.2, igraph_1.0.166) to determine families of SMGC clusters. Finally, we ran another round of random walk clustering on the communities that contained more members than species in the analysis (ptailoring/pbackbone = sum of percentage BLAST alignment of tailoring/backbone enzymes, respectively; ntailoring/nbackbone = number of tailoring/backbone enzymes with significant hits, respectively; ttailoring/tbackbone = total number of tailoring/backbone enzymes):

$$\mathrm{ptailoring} \times \frac{\mathrm{ntailoring}}{\mathrm{ttailoring}} \times 0.35 + \mathrm{pbackbone} \times \frac{\mathrm{nbackbone}}{\mathrm{tbackbone}} \times 0.65 \qquad (1)$$

To create a cluster similarity score, a combined score of tailoring and backbone enzymes was created. The sum of the BLASTp percent identity (ptailoring/pbackbone) of all hits for tailoring enzymes between two clusters was divided by the maximum amount of tailoring enzyme (ttailoring/tbackbone) and multiplied by 0.35. Then the score for the backbone enzymes was calculated in the same manner but multiplied by 0.65 to give more weight to the backbone enzymes. The scores were added to create an overall cluster similarity score:

$$\mathrm{avg}\left(\mathrm{pident}_{\mathrm{tailoring}}\right) \times 0.35 + \mathrm{avg}(\mathrm{pident}_{\mathrm{backbones}}) \times 0.65 \qquad (2)$$

**Annotation of SMGC families using MIBiG (genetic dereplication)**. SMGC families were annotated based on the MIBiG database[67]. Known gene clusters were coupled to SMGC families, making it possible to predict the compounds or derivatives thereof a species can potentially produce. Cluster families containing one cluster highly similar to a known compound cluster are labeled after the known compound.

**Genome synteny analysis**. Orthologs were defined as a pair of genes found between two genomes from different species by bidirectional best hits using BLASTP with e-value < 10¹⁰. When two genes within 10 kbp on the first genome have corresponding orthologs within 10 kbp on the second genome, the region between the two genes was defined as a syntenic block. The distance between the two genes was calculated by the formula, |PC1 – PC2| – 1/2 (LN1 + LN2), where PCn and LNn are nucleotide position of the center and nucleotide length of gene "n" (n = 1 or 2), respectively.

**Analysis of chromosomal localization**. For visualization of chromosome, chromosomal location, and gene density the R package karyoploteR was used[85].

**Secondary metabolite gene cluster analysis and visualization**. For visualization of cluster synteny and similarity EasyFig was used[86]. The parameters minimum length and minimum identity were set to 50 bp and 50%, respectively.

**Profiling of growth on different carbon sources**. The species were grown on 35 different media plates using the same method as first described by deVries et al.[55] All strains were grown on MM[87] containing monosaccharides/oligosaccharides, polysaccharides, and crude substrates at 25 mM, 1%, and 3% final concentration, respectively.

**Chemical analysis of secondary metabolism**. The section *Flavi* strains were cultivated as three-point cultures on CYA and YES media for 7 days in the dark at 30 °C; subsequently three plugs (6 mm inner diameter) were taken across the colony, 800 μL of isopropanol ethyl acetate (1:3 v/v) with 1% formic acid was added and ultrasonicated for 1 h. The liquid sample was transferred to another tube and evaporated; after this 300 μL of methanol was added to dissolve the pellets and the samples were ultrasonicated for 20 min[88–90]. Samples were then centrifuged at max g-power for 2–3 min, and afterward 150 μL of the supernatant was transferred to HPLC vials[91–97].

Ultra-high-performance liquid chromatography–diode array detection–quadrupole time-of-flight mass spectrometry (UHPLC–DAD–QTOFMS) was performed on an Agilent Infinity 1290 UHPLC system equipped with a diode array detector. Separation was obtained on a 250 × 2.1 mm i.d., 2.7 μm, Poroshell 120 Phenyl Hexyl column (Agilent Technologies, Santa Clara, CA) held at 60 °C. The sample, 1 μL, was eluted with a flow rate of 0.35 mL min⁻¹ using A: a linear gradient 10% acetonitrile in Milli-Q water buffered with 20 mM formic acid increased to 100% in 15 min, staying there for 2 min before returning to 10% in 0.1 min, held for 3 min before the following run.

Mass spectrometry (MS) detection was performed on an Agilent 6545 QTOF MS equipped with an Agilent dual-jet stream electro spray ion (ESI) source with a drying gas temperature of 160 °C, gas flow of 13 L min⁻¹, sheath gas temperature of 300 °C, and flow of 16 L min⁻¹. Capillary voltage was set to 4000 V, and nozzle voltage, to 500 V in positive mode. MS spectra were recorded as centroid data, at an m/z of 100–1700, and auto MS/HRMS fragmentation was performed at three collision energies (10, 20, and 40 eV), on the three most intense precursor peaks per cycle. The acquisition was 10 spectra s⁻¹. Data were handled in the software Agilent MassHunter Qualitative Analysis (Agilent Technologies, Santa Clara, CA).

**Reporting summary**. Further information on research design is available in the Nature Research Reporting Summary linked to this article.

## Data availability

All genomes used in the study are available from Joint Genome Institute fungal genome portal MycoCosm (http://jgi.doe.gov/fungi). All new genomes published in the study have been deposited in GenBank under the following NCBI accession numbers: Aspergillus nomius IBT 12657—BioProject: PRJNA333904, Accession No: SWDY00000000; Aspergillus minisclerotigenes CBS 117635—BioProject: PRJNA333902, Accession No: SWDZ00000000; Aspergillus parasiticus CBS 117618—BioProject: PRJNA333906, Accession No: SWCZ00000000; Aspergillus novoparasiticus CBS 126849 —BioProject: PRJNA333905, Accession No: SWDA00000000; Aspergillus leporis CBS 151.66—BioProject: PRJNA333901, Accession No: SWBU00000000; Aspergillus alliaceus CBS 536.65—BioProject: PRJNA334014, Accession No: SWAS00000000; Aspergillus parvisclerotigenus CBS 121.62—BioProject: PRJNA333907, Accession No: SWAT00000000; Aspergillus pseudotamarii CBS 117625— BioProject: PRJNA333910, Accession No: STFH00000000; Aspergillus avenaceus IBT 18842— BioProject: PRJNA333897, Accession No: STFI00000000; Aspergillus tamarii CBS 117626— BioProject: PRJNA333912, Accession No: STFJ00000000; Aspergillus transmontanensis CBS 130015—BioProject: PRJNA333913, Accession No: STFK00000000; Aspergillus sergii CBS 130017—BioProject: PRJNA333911, Accession No: STFL00000000; Aspergillus arachidicola CBS 117612—BioProject: PRJNA333896, Accession No: STFM00000000; Aspergillus coremiiformis CBS 553.77—BioProject: PRJNA333900, Accession No: STFN00000000; Aspergillus caelatus CBS 763.97—BioProject: PRJNA333899, Accession No: STFO00000000; Aspergillus bertholletius IBT 29228— BioProject: PRJNA333898, Accession No: STFP00000000; Aspergillus albertensis IBT 14317—BioProject: PRJNA333895, Accession No: STFQ00000000; Aspergillus pseudonomius CBS 119388—BioProject: PRJNA333909, Accession No: STFR00000000; Aspergillus pseudocaelatus CBS 117616—BioProject: PRJNA333908, Accession No: STFS00000000.

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

## Acknowledgements
Genome sequencing was kindly supported by Joint BioEnergy Institute and Joint Genome Institute. M.R.A., J.L.N., S.T. and T.C.V. gratefully acknowledge funding from the Villum Foundation, Grant VKR023437. M.R.A. and T.C.V. further acknowledge funding from the Danish National Research Foundation, grant number DNRF137. M.R.A. further acknowledges support from the Jorck Foundation. The work conducted by the US Department of Energy Joint Genome Institute, a US Department of Energy Office of Science User Facility, is supported by the Office of Science of the US Department of Energy under Contract No. DE-AC02-05CH11231. The US Department of Energy Joint BioEnergy Institute (www.jbei.org) is supported by the US Department of Energy, Office of Science, and Office of Biological and Environmental Research, through Contract DE-AC02-05CH11231 between Lawrence Berkeley National Laboratory and the US Department of Energy.

## Author contributions
I.K. analyzed data, contrived data analysis methods, contributed to design of research, and wrote most parts of the paper. T.V. conceived the overall project, analyzed data, contributed to design of research, wrote parts of the paper, and coordinated the project. J.C.F. contributed to design of research, contributed analytical tools and data for species selection and verification, wrote parts of the paper, and analyzed data. J.L.N. and S.T. analyzed data and contributed to design of research. S.K. and T.I. generated data on secondary metabolism and analyzed chemical data. E.K.L. and M.E.K. contributed to design of research, developed methods, conducted experiments, and analyzed data. A. Sat analyzed data and contributed to design of research. A.W., R.S.K. and R.J.M.L. performed part of the experiments. M.R.M. analyzed data and wrote parts of the paper. A.K., A. Sal, S.H., R.R. and S.M. annotated genomes and analyzed data. A.C., A.L., K.L., and J.P. assembled the genomes. C.D., G.H. and M.C. sequenced RNA and DNA. J.K.M. and B.A.S. contributed to design of research. B.H. and E.D. contributed analytical tools and analyzed CAZyme data. U.H.M. contributed to design of research and developed methods. T.O.L. generated data on secondary metabolism, analyzed data, and wrote parts of the paper. R.P.dV. analyzed data and wrote parts of the paper. K.B. and I.G. coordinated the DNA and RNA sequencing and annotation. M.M. analyzed data and wrote parts of the paper. S.E.B. conceived the overall project, analyzed data, contributed to design of research, and contributed to writing and editing the paper. M.R.A. conceived the overall project, analyzed data, contributed to design of research, wrote parts of the paper, and coordinated the project. All authors commented on the paper.

## Competing interests
The authors declare no competing interests.
