## [Peer Review File · Nature Communications]

Reviewers' comments:

Reviewer #1 (Remarks to the Author):

The authors of manuscript NCOMMS-19-11952-T are reporting the de novo sequencing of the genomes of 19 formerly unsequenced members of the *Aspergillus* taxonomic section Flavi. The Flavi section is of particular importance because it contains species used in economically important food fermentation processes, it contains notorious economically important toxin producing strains, and strains important in emerging biotechnology applications. It also contains a major human aspergillosis pathogen that infects immunocompromised human patients. All the newly sequence strains were annotated using a single annotation pipeline, the US Department of Energy Joint Genome Institute fungal genome annotation pipeline. This standardization of the annotation process greatly supports the credibility of the comparative genomic analysis reported in the manuscript. Comparative genome analysis was reported using the newly sequenced strains, additional sequenced strains from the Flavi section, and additional notable *Aspergillus* strains, a *Penicillium*, and a *Neurospora* to total 31 fungal genomes.

The manuscript is very well written, the figures are very clear and informative. The authors report on the findings of genomic comparisons across these strains in several targeted categories including the quality measures of the genome sequence of each strain, features of the section Flavi genomes such as genome size and evolutionary heritage, and diversity in selected categories of genes. These genes were selected for particularly important categories of genes related to important phenotypes. They included secondary metabolite biosynthetic gene clusters and associated chromosome organization, species-specific genes involved in regulation and in P450 activities, and in carbohydrate-active enzymes. Coupled with these genomic comparisons was reports of some limited phenotypic analyses related to the gene categories compared.

I do have a couple of minor suggestions to offer for improvement of the manuscript. The first is that a preliminary view of some of the genomic organizational findings was reported in a 2008 publication of a similar study of a much more limited number of *Aspergillus* strains and species that would perhaps be appropriately cited. The citation is:

Fedorova, N.D., Joardar, V., Crabtree, J., Anderson, M., Maiti, R., Amedeo, P., Wortman, J.R., Albarraq, A., Angiuoli, S., Bowyer, P., Bussey, H., Cyer, P., Egan, A., Galens, A., Haas, B., Inman, J., Lemieux, S., Malavazi, I., Orvis, J., Roemer, T., Ronning, C., Silva, J., Sundaram, J., Whitty, B., Youngman, P., Fraser, C.M., Sutton, G., Venter, J.C., White, W., Goldman, G., Turner, G., Jiang, B., Denning, D., Nierman, W.C. 2008. Genomic islands in the pathogenic filamentous fungus *Aspergillus fumigatus*. *PLoS Genetics*, 4:e1000046. PMID18404212.

The second suggestion is more subtle. In Figure 1 the quality statistics are provided for the reported fungal genomes. The column showing the number of scaffolds in the genome assemblies indicates several with fewer than 8 scaffolds in the genome assembly. It has been only very recently that these low numbers of scaffolds can be attained using only assembled sequence reads via single molecule long-read technologies such as Oxford Nanopore. For many genomes in this table the chromosome size scaffolds such as those of *Aspergillus oryzae* RIB40 shown in Figure 3 and *Aspergillus fumigatus* Af293 were achieved by combining sequence read assembly scaffolds with restriction Optical Maps as described in the original cited genome papers (Machida et al. 2005 and Nierman et al. 2005). For the sense of the history of sequencing technology employed from 2005 to the present reflected in the studied genomes, it might be worth a brief mention of the Optical Mapping technology employed in the earliest sequencing of *Aspergillus* genomes.

The manuscript will serve as an important reference for academic, industrial, and medical *Aspergillus* researchers for years to come. Because of the high quality of the work, the clear and well written language and figures of the manuscript and the long-term reference value of the data and findings I recommend it for publication in *Nature Communications*.

William C. Nierman

Reviewer #2 (Remarks to the Author):

Review: Fungal friends and foes – A comparative genomics study of 23 *Aspergillus* species from section Flavi

In this manuscript, the authors provide genome assemblies for 19 previously unsequenced *Aspergillus* species in section Flavi, explore phylogenetic relationships between these new genomes and previously available ones, and compare gene content. The gene content comparison includes secondary metabolite gene clusters (SMGCs) and carbohydrate-active enzymes (CAZymes), both classes of genes / pathways whose members are important for pathogenicity and enzymatic activity of fungal species. The manuscript presents novel data and analyses of genome sequencing efforts for species within *Aspergillus*, which will be useful to the scientific community, particularly in industrial and agricultural settings. The science behind the manuscript is sound, with a variety of methods to support conclusions.

Major revisions:

The results section is quite thorough, although the authors tend to add discussion into this section. In contrast, the discussion section is very short and largely repeats the authors' findings rather than adds discussion. The authors could either merge results & discussion or alternatively remove the various discussion-style sentences in their results and write an informative discussion. Either way, we strongly encourage the authors to expand the discussion and place their results in the context of previous literature on the topic.

Within the methods section, the authors cite previous work rather than describe the processes they used, limiting the usefulness of the section for readers who would like to use similar methods in their own work. The methods section would be more useful with brief explanations of the steps taken, such that readers could replicate the study without searching through multiple other references which may be behind paywalls. The subsections most in need of additional explanation are:

- 1) DNA and RNA preparation, sequencing, and assembly
- 2) Genome annotation
- 3) Prediction of Carbohydrate-Active Enzymes
- 4) Prediction of secondary metabolite gene clusters
- 5) Profiling of growth on different carbon sources

title: "Fungal friends and foes" - this could be said of virtually every group of fungi (e.g., it could be said of *Aspergillus* in general, of *Aspergillaceae*, of *Eurotiomycetes*, etc.). Remove?

Minor revisions:

Line 34: change "genetic variations" to "variations in gene content" (genetic variation typically refers to SNPs and other types of polymorphisms within a species)

Lines 70-72: check grammar of this sentence

Line 74: add "to" after "belong"

Line 78: here, it would be good to also state the total number of known species in section Flavi

Line 86: Figure 1 panel B is presented before 1A

Line 94: what are "monocore" genes?

Figure 1: note that the phylogeny is rooted incorrectly. My understanding is that the correct tree is: ((*Aspergillus*, *Penicillium*), *Neurospora*) – i.e., *Penicillium* is an outgroup of all *Aspergillus* taxa and *Neurospora* is an outgroup of *Aspergillus*+*Penicillium*

Lines 112-114 of the results: authors reference a 2005 paper and state that "*A. oryzae* has previously been reported to have a larger genome than other *Aspergilli* (27) and the data presented here suggests this is a trait shared by most species in section Flavi." As of 2005, very few *Aspergilli* had sequenced genomes. Consider changing this wording to "Most section Flavi

species have large genomes compared to other *Aspergillus* sections, as was previously reported for *A. oryzae*."

Line 124: remove comma between "is" and "that"

Lines 139-143, which correspond to Figure 2: the numbering and italics are unnecessary. These bullet points could be written as sentences.

Line 170: change "as" to "although"

Line 184: genome should not be italicized

Line 249: change "genetic diversity" with "variation in gene content"

Line 450: make gene plural

Line 454: add "the" before "PROTGAMMAWAG substitution model"

Reviewer #3 (Remarks to the Author):

Kjærboelling et al. de novo sequenced 19 genomes of *Aspergillus* species in the section of Flavi. By comparisons of 31 fungal genomes, including 23 section Flavi species and 8 other *Aspergillus*, *Penicillium*, and *Neurospora* species, the authors have shown various aspects of genome diversity of this section. Further analysis of carbohydrate-active enzymes (CAZymes) and secondary metabolism gene clusters (SMGCs) offers substantial open resources for the fungal research community.

A few suggestions that could improve the manuscript are listed below.

1. Line 43: "20 types of compounds across 31 species" while Figure 6 only lists 29 species.
2. Figure 1: The zoom in the box has a different bootstrap value from the *A. oryzae* branch, 91 vs 94.
3. Figure 2, line 146: "phylogenetic relationship between the 31 *Aspergilli*" while Figure 2 only lists 29 *Aspergilli*.
4. Line 331: "This is more than 20 extra per species". However, based on reference 70, an average of 54.9 putative clusters per *Penicillium* species was identified. Therefore, that's 18 extra clusters per species compared to the *Penicillium* genus.
5. Line 349: Supplementary Table 4 contains little information as the numbers of clusterFam do not provide any concept of the type of metabolites it could potentially produce. A more comprehensive Table includes 1) the size of the backbone enzyme, 2) the class of the backbone enzyme, and 3) the domain organization (if the backbone enzyme is a PKS, NRPS, or PKS-NRPS) will provide more information for linking secondary metabolites to gene clusters. Also, the authors use *A. parvisclerotigenus* instead of *A. aflatoxiformans* in Supplementary Table 4 while *A. aflatoxiformans* was used throughout the manuscript. Does this Table include terpene cyclase (TC) backbone enzyme?
6. Line 351: "unique clusters in each species (6.8 unique SMGCs/species)". Should this number be 5.9 since 130 unique clusters were found in 22 species (Figure 6)?
7. Line 377-383: By combining chemical analysis and retro-biosynthesis analysis from the chemical structure of miyakamides, the authors were able to propose the likely genes involved in the biosynthesis of miyakamides. As mentioned in point 5, Supplementary Table 4 currently does not provide enough information that allows readers to link the miyakamide biosynthesis cluster to clusterFam 31. It can be quickly visualized if the domain organization of all NRPSs has been provided in Supplementary Table 4. This is because not many NRPSs contain an N-methyltransferase (MT) domain. Furthermore, as shown in Supplementary Figure 10B, the synteny plot of genes for the predicted miyakamide biosynthesis is well conserved, suggesting that the miyakamide NRPSs in different species have similar domain organizations. A close inspection of the miyakamide NRPSs listed in Supplementary Figure 10B from the JGI website indicated that all miyakamide NRPSs share the C*-A-T-C-A-MT-T-C-A-T-TE domain organization with ~3900 amino

acids. The only exception is the NRPS from *A. novoparasiticus* which has a T-C-A-MT-T-C-A-T-TE domain organization with ~2900 amino acids. The missing ~1000 amino acids with the C*-A domains in *A. novoparasiticus* is likely due to the mis-annotation of the start codon. If this is the case, all miyakamide NRPSs contain three A domains with a MT domain in the second module. Authors should double check the domain organization of the miyakamide NRPSs cause, currently in the manuscript, two A domains were reported in three *Aspergillus* species (Line 384).

8. Line 528: "and analyzed data chemical data", remove first "data".

9. Reference 3 and 20, 13 and 20, 14 and 22, are the same. Also, italicize the species name in the Reference section.

Dear reviewers

We are very happy to see the positive comments and with the few additional changes based on the reviewers' excellent suggestions we think this manuscript has become even better and will serve as an important reference article in the future. Below are the reviewers' comments and our response in green.

Reviewers' comments:

Reviewer #1 (Remarks to the Author):

The authors of manuscript NCOMMS-19-11952-T are reporting the de novo sequencing of the genomes of 19 formerly unsequenced members of the *Aspergillus* taxonomic section Flavi. The Flavi section is of particular importance because it contains species used in economically important food fermentation processes, it contains notorious economically important toxin producing strains, and strains important in emerging biotechnology applications. It also contains a major human aspergillosis pathogen that infects immunocompromised human patients. All the newly sequence strains were annotated using a single annotation pipeline, the US Department of Energy Joint Genome Institute fungal genome annotation pipeline. This standardization of the annotation process greatly supports the credibility of the comparative genomic analysis reported in the manuscript. Comparative genome analysis was reported using the newly sequenced strains, additional sequenced strains from the Flavi section, and additional notable *Aspergillus* strains, a *Penicillium*, and a *Neurospora* to total 31 fungal genomes.

The manuscript is very well written, the figures are very clear and informative. The authors report on the findings of genomic comparisons across these strains in several targeted categories including the quality measures of the genome sequence of each strain, features of the section Flavi genomes such as genome size and evolutionary heritage, and diversity in selected categories of genes. These genes were selected for particularly important categories of genes related to important phenotypes. They included secondary metabolite biosynthetic gene clusters and associated chromosome organization, species-specific genes involved in regulation and in P450 activities, and in carbohydrate-active enzymes. Coupled with these genomic comparisons was reports of some limited phenotypic analyses related to the gene categories compared.

I do have a couple of minor suggestions to offer for improvement of the manuscript.

The first is that a preliminary view of some of the genomic organizational findings was reported in a 2008 publication of a similar study of a much more limited number of *Aspergillus* strains and species that would perhaps be appropriately cited. The citation is:

Fedorova, N.D., Joardar, V., Crabtree, J., Anderson, M., Maiti, R., Amedeo, P., Wortman, J.R., Albarraq, A., Angiuoli, S., Bowyer, P., Bussey, H., Cyer, P., Egan, A., Galens, A., Haas, B., Inman, J., Lemieux, S., Malavazi, I., Orvis, J., Roemer, T., Ronning, C., Silva, J., Sundaram, J., Whitty, B., Youngman, P., Fraser, C.M., Sutton, G., Venter, J.C., White, W., Goldman, G., Turner, G., Jiang, B., Denning, D., Nierman, W.C. 2008. Genomic islands in the pathogenic filamentous fungus *Aspergillus fumigatus*. *PLoS Genetics*, 4:e1000046. PMID18404212.

This is an excellent point and we have added the reference at Line 180.

The second suggestion is more subtle. In Figure 1 the quality statistics are provided for the reported fungal genomes. The column showing the number of scaffolds in the genome assemblies indicates several with fewer than 8 scaffolds in the genome assembly. It has been only very recently that

these low numbers of scaffolds can be attained using only assembled sequence reads via single molecule long-read technologies such as Oxford Nanopore. For many genomes in this table the chromosome size scaffolds such as those of *Aspergillus oryzae* RIB40 shown in Figure 3 and *Aspergillus fumigatus* Af293 were achieved by combining sequence read assembly scaffolds with restriction Optical Maps as described in the original cited genome papers (Machida et al. 2005 and Nierman et al. 2005). For the sense of the history of sequencing technology employed from 2005 to the present reflected in the studied genomes, it might be worth a brief mention of the Optical Mapping technology employed in the earliest sequencing of *Aspergillus* genomes.

This is a good point and we have added a sentence in the figure legend of figure 1.

The manuscript will serve as an important reference for academic, industrial, and medical *Aspergillus* researchers for years to come. Because of the high quality of the work, the clear and well written language and figures of the manuscript and the long-term reference value of the data and findings I recommend it for publication in Nature Communications.

William C. Nierman

Reviewer #2 (Remarks to the Author):

Review: Fungal friends and foes – A comparative genomics study of 23 *Aspergillus* species from section Flavi

In this manuscript, the authors provide genome assemblies for 19 previously unsequenced *Aspergillus* species in section Flavi, explore phylogenetic relationships between these new genomes and previously available ones, and compare gene content. The gene content comparison includes secondary metabolite gene clusters (SMGCs) and carbohydrate-active enzymes (CAZymes), both classes of genes / pathways whose members are important for pathogenicity and enzymatic activity of fungal species. The manuscript presents novel data and analyses of genome sequencing efforts for species within *Aspergillus*, which will be useful to the scientific community, particularly in industrial and agricultural settings. The science behind the manuscript is sound, with a variety of methods to support conclusions.

Major revisions:

The results section is quite thorough, although the authors tend to add discussion into this section. In contrast, the discussion section is very short and largely repeats the authors' findings rather than adds discussion. The authors could either merge results & discussion or alternatively remove the various discussion-style sentences in their results and write an informative discussion. Either way, we strongly encourage the authors to expand the discussion and place their results in the context of previous literature on the topic.

Thank you for this point. We have merged the results and discussions sections as suggested by the reviewer, converted the previous Discussions section into a Conclusions section, and included literature as suggested by the reviewers.

Within the methods section, the authors cite previous work rather than describe the processes they

used, limiting the usefulness of the section for readers who would like to use similar methods in their own work. The methods section would be more useful with brief explanations of the steps taken, such that readers could replicate the study without searching through multiple other references which may be behind paywalls. The subsections most in need of additional explanation are:

- 1) DNA and RNA preparation, sequencing, and assembly
- 2) Genome annotation
- 3) Prediction of Carbohydrate-Active Enzymes
- 4) Prediction of secondary metabolite gene clusters
- 5) Profiling of growth on different carbon sources

The method section has been updated with thorough explanations instead of the style with references.

title: "Fungal friends and foes" - this could be said of virtually every group of fungi (e.g., it could be said of *Aspergillus* in general, of *Aspergillaceae*, of *Eurotiomycetes*, etc.). Remove?

Minor revisions:

Line 34: change "genetic variations" to "variations in gene content" (genetic variation typically refers to SNPs and other types of polymorphisms within a species)

Has been updated as suggested

Lines 70-72: check grammar of this sentence

The sentence has been updated to:

Given the importance of section *Flavi*, it is highly valuable to examine the full genetic potential of the section in order to assess alternative species for industrial use, combat pathogenicity, find novel bioactives and to identify useful enzymes.

Line 74: add "to" after "belong"

Done

Line 78: here, it would be good to also state the total number of known species in section *Flavi*

Added: containing at least 29 species.

Line 86: Figure 1 panel B is presented before 1A

We believe the results are presented in the most logical order and would like to have panel A before B but it is not possible to change the order those panels in the figure without lowering the readability of the figure. We have therefore kept this part but since it is clearly stated what part of the figure we are referring to in the text we believe it is easy to follow.

Line 94: what are "monocore" genes?

Added explanation: (a single homolog in each of the species)

Figure 1: note that the phylogeny is rooted incorrectly. My understanding is that the correct tree is: ((*Aspergillus*, *Penicillium*), *Neurospora*) – i.e., *Penicillium* is an outgroup of all *Aspergillus* taxa and *Neurospora* is an outgroup of *Aspergillus*+*Penicillium*

This is correct. We have used the taxonomically incorrect rooting to emphasize the *Aspergillus*-clade in the tree. We have removed the rooting to avoid confusion.

Lines 112-114 of the results: authors reference a 2005 paper and state that "*A. oryzae* has previously been reported to have a larger genome than other *Aspergilli* (27) and the data presented here suggests this is a trait shared by most species in section *Flavi*." As of 2005, very few *Aspergilli*

had sequenced genomes. Consider changing this wording to “Most section *Flavi* species have large genomes compared to other *Aspergillus* sections, as was previously reported for *A. oryzae*.”

This is a good point and we have changed the section to:

The genome sizes of *Aspergillus* section *Flavi* are generally large compared to other representative Aspergilli (average of 37.96 Mbp vs 31.7 Mbp (Figure 1C)), as was previously reported for *A. oryzae*²⁷

Line 124: remove comma between “is” and “that”

The comma has been removed

Lines 139-143, which correspond to Figure 2: the numbering and italics are unnecessary. These bullet points could be written as sentences.

We see the point, but prefer this version for clarity.

Line 170: change “as” to “although”

We don’t think ‘although’ gives the intended meaning but have changed the sentence to the following to make the intended meaning clearer:

We will focus on InterPro since it covers more genes: The most common InterPro functions

Line 184: genome should not be italicized

Has been corrected.

Line 249: change “genetic diversity” with “variation in gene content”

Has been changed as suggested.

Line 450: make gene plural

Has been corrected

Line 454: add “the” before “PROTGAMMAWAG substitution model”

A ‘the’ has been added.

Reviewer #3 (Remarks to the Author):

Kjærboelling et al. de novo sequenced 19 genomes of *Aspergillus* species in the section of *Flavi*. By comparisons of 31 fungal genomes, including 23 section *Flavi* species and 8 other *Aspergillus*, *Penicillium*, and *Neurospora* species, the authors have shown various aspects of genome diversity of this section. Further analysis of carbohydrate-active enzymes (CAZymes) and secondary metabolism gene clusters (SMGCs) offers substantial open resources for the fungal research community.

A few suggestions that could improve the manuscript are listed below.

1. Line 43: “20 types of compounds across 31 species” while Figure 6 only lists 29 species.

Has been updated to reflect the figure.

2. Figure 1: The zoom in the box has a different bootstrap value from the *A. oryzae* branch, 91 vs 94.

The zoom box of the figure has now been updated to show the correct number

3. Figure 2, line 146: “phylogenetic relationship between the 31 *Aspergilli*” while Figure 2 only lists 29 *Aspergilli*.

Has been changed to 29.

4. Line 331: “This is more than 20 extra per species”. However, based on reference 70, an average of 54.9 putative clusters per *Penicillium* species was identified. Therefore, that’s 18 extra clusters per species compared to the *Penicillium* genus.

The sentence has been changed to: This is more than 15 extra per species

5. Line 349: Supplementary Table 4 contains little information as the numbers of clusterFam do not provide any concept of the type of metabolites it could potentially produce. A more comprehensive Table includes 1) the size of the backbone enzyme, 2) the class of the backbone enzyme, and 3) the domain organization (if the backbone enzyme is a PKS, NRPS, or PKS-NRPS) will provide more information for linking secondary metabolites to gene clusters. Also, the authors use *A. parvisclerotigenus* instead of *A. aflatoxiformans* in Supplementary Table 4 while *A. aflatoxiformans* was used throughout the manuscript. Does this Table include terpene cyclase (TC) backbone enzyme?

This is a good point, we have updated the table to include more information. The table does not include Terpene cyclases, as they are very tricky to predict automatically. Using the protein ID, more information can be obtained from the JGI website for specific purposes.

6. Line 351: “unique clusters in each species (6.8 unique SMGCs/species)”. Should this number be 5.9 since 130 unique clusters were found in 22 species (Figure 6)?

The number of 130 was an error instead it should be 150, and the 6.8 unique SMGCs/species is correct. We have changed 130 to 150 in the text.

7. Line 377-383: By combining chemical analysis and retro-biosynthesis analysis from the chemical structure of miyakamides, the authors were able to propose the likely genes involved in the biosynthesis of miyakamides. As mentioned in point 5, Supplementary Table 4 currently does not provide enough information that allows readers to link the miyakamide biosynthesis cluster to clusterFam 31. It can be quickly visualized if the domain organization of all NRPSs has been provided in Supplementary Table 4. This is because not many NRPSs contain an N-methyltransferase (MT) domain. Furthermore, as shown in Supplementary Figure 10B, the synteny plot of genes for the predicted miyakamide biosynthesis is well conserved, suggesting that the miyakamide NRPSs in different species have similar domain organizations. A close inspection of the miyakamide NRPSs listed in Supplementary Figure 10B from the JGI website indicated that all miyakamide NRPSs share the C*-A-T-C-A-MT-T-C-A-T-TE domain organization with ~3900 amino acids. The only exception is the NRPS from *A. novoparasiticus* which has a T-C-A-MT-T-C-A-T-TE domain organization with ~2900 amino acids. The missing ~1000 amino acids with the C*-A domains in *A. novoparasiticus* is likely due to the mis-annotation of the start codon. If this is the case, all miyakamide NRPSs contain three A domains with a MT domain in the second module. Authors should double check the domain organization of the miyakamide NRPSs cause, currently in the manuscript, two A domains were reported in three *Aspergillus* species (Line 384).

For the first part we have updated Table 4 to include the backbone type and predicted InterPro domains.

For the second part concerning the domain structure and annotation we agree that it is likely due to mis-annotation and that they all most likely contain three A domains and a MT domain. We have added a sentence stating this.

8. Line 528: “and analyzed data chemical data”, remove first “data”.

The first ‘data’ has been removed.

9. Reference 3 and 20, 13 and 20, 14 and 22, are the same. Also, italicize the species name in the Reference section.

The names of species have been italicized in the reference section and the additional copy of the references that had two copies have been deleted.